# Surfside Science

Sean-David Brokke, Amaryllis Lee, Anthony Sevold, Manuel Rojas

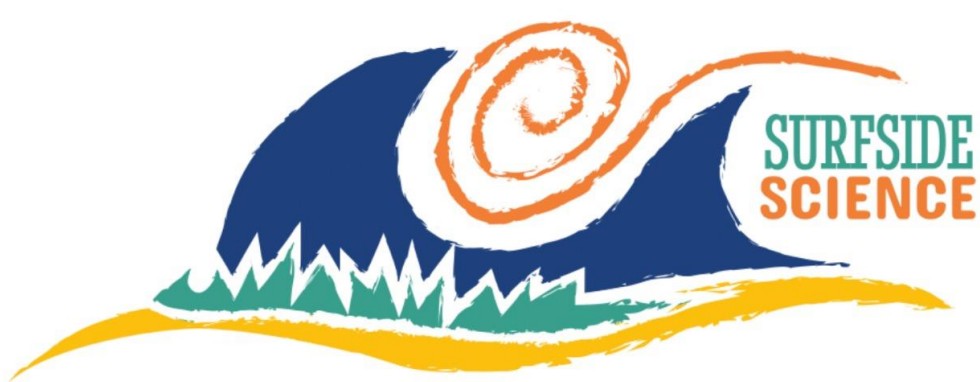

## Abstract

The goal of the project is to develop and validate methodologies that enhance accessibility to data collection systems for Small Island Developing States (SIDS), primarily by creating affordable and accurate environmental monitoring prototypes and making them open-source, accessible to the public through our project website. These prototypes are designed to be replicable, even by individuals and communities where environmental monitoring efforts are currently limited. The aim is to empower people to become environmental scientists, providing them with the tools and knowledge to monitor and address climate and environmental issues in their regions.

Many islands in the Caribbean and other SIDS are not part of the EU climate agreement. This highlights the critical importance of climate monitoring in the Caribbean. These islands are highly vulnerable to the adverse effects of climate change, and the data is essential for understanding their unique environmental challenges. Without accurate Caribbean-specific data, future climate solutions may not effectively address the specific needs of this region, which possesses a distinct and vulnerable environment.

The Surfside Science project has established monitoring methods to measure critical environmental factors, such as air quality, water quality, coastline monitoring, and seafloor mapping. These methods provide real-time, reliable data and have been operational for over a year, affirming the viability of our solution. Users can access live environmental data for Aruba through our data portal, and our method validations confirm data reliability.

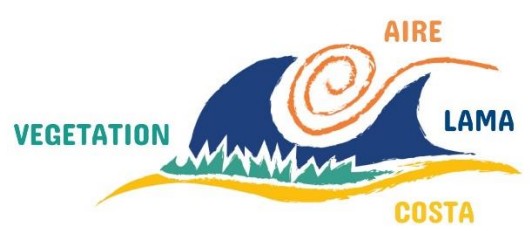

# 1 Introduction

The Surfside Science project represents a pioneering endeavor dedicated to advancing the monitoring of coastal and marine ecosystems, with a particular focus on SIDS. Our primary goal is to address the challenge of limited accessibility to reliable environmental data collection in regions like Aruba.

The problem Surfside Science aims to address is the lack of accessible and environmental monitoring solutions, particularly for SIDS. In these regions, critical environmental data, such as air and water quality, is often scarce due to the limitations in resources and technology. This gap in data collection inhibits the ability to understand and mitigate environmental challenges, especially in the face of climate change.

This problem is of paramount importance due to its broad implications for environmental conservation and public health. SIDS, like Aruba, are particularly vulnerable to the adverse effects of climate change and pollution. Reliable environmental data is essential for creating effective strategies to address these issues. Furthermore, our solution serves a global need as it provides an accessible model for data collection, empowering not only Aruba but also similar regions worldwide.

Our solution is to develop replicable and cost-effective environmental monitoring prototypes, including air and water quality sensors, seafloor mapping techniques, and satellite-based coastal change analysis. These prototypes are designed to be open-source, user-friendly, and widely accessible. By making our methodologies and prototypes openly available, we enable individuals and communities, especially those in regions with limited climate monitoring, to initiate their own data collection initiatives.

The Surfside Science Project offers an innovative solution through the implementation of accessible, low-cost, and replicable environmental monitoring techniques. By installing monitoring stations for physical parameters, air quality, and marine environment, and by developing a marine mapping system based on satellite imagery and underwater photography, the project creates a comprehensive, data-driven ecosystem. Open-access data, publicly shared through an online database and web portal, enhances community participation, policy development, and scientific research. This open approach ensures that the problem of data inaccessibility is overcome, fostering community involvement, and increased environmental awareness.

The results of this project include the installation of monitoring stations measuring critical environmental parameters. These stations, which measure air quality, water acidity, temperature, dissolved oxygen, and particulate matter, will provide real-time, reliable data. The Surfside Science project has successfully deployed a comprehensive network of air quality sensors, including PM10.0, PM2.5, and PM1 measurements.

The project also involves the creation of an integrated marine mapping system, which combines satellite imagery classification, underwater photography analysis, and field validation to assess key aspects of the marine study area. Additionally, the project will generate validated baseline GIS data for analyzing land cover change, coastal erosion, and shallow reef coverage.

Our project aligns with key Sustainable Development Goals (SDGs), including SDG 13 (Climate Action) and SDG 14 (Life Below Water), emphasizing our commitment to addressing global environmental challenges. Through open-source knowledge sharing and user-friendly technical instructions, we provide a tangible model for empowering communities to become environmental activists and drive positive change in their regions.

These sensors have operated efficiently for over a year, demonstrating the viability of our solution. We also have a data portal where users can access our live environmental data of Aruba. Furthermore, our validation methods have been applied to assess data reliability.

## 2 Goals

1. **Five aquatic monitoring stations were installed measuring acidity, temperature, and dissolved oxygen around the Surfside Marina Strip in Aruba:**
   - **Rationale:** These monitoring stations serve as the foundation for data collection, ensuring real-time data on critical water parameters.
2. **Five air quality monitoring stations installed measuring particulate matter, temperature, and humidity at sea level around the Surfside Marina Strip in Aruba:**
   - **Rationale:** Monitoring air quality contributes to a comprehensive understanding of environmental factors affecting Aruba's coastal and marine ecosystems.
3. **Integrated marine mapping system combining satellite imagery classification, underwater photography analysis, and field validation:**
   - **Rationale:** The mapping system provides a multidimensional view of the marine environment, allowing researchers to analyze and interpret key parameters.
4. **Validated baseline GIS data for studying land cover change, coastal erosion, and shallow reef coverage:**
   - **Rationale:** Access to baseline data supports long-term research and helps assess the impact of environmental changes.
5. **Online database and web portal for openly accessing collected data and data products:**
   - **Rationale:** An open-access platform ensures that data and results are readily available to the public, researchers, and policymakers.
6. **Replicable scientific methods for monitoring marine ecosystem health, clearly documented, including validated protocols and code for automation of data processing:**
   - **Rationale:** Sharing replicable methods and clear documentation encourages wider adoption and empowers others to contribute to environmental monitoring.
7. **Report summarizing the status of the coastal and marine environment at Surfside Marine Strip in Aruba:**
   - **Rationale:** The report compiles findings, data, and recommendations, facilitating knowledge dissemination and informing future research and conservation efforts.

The Surfside Science Project, through its well-defined goals, objectives, and specific outputs, aims to strengthen Aruba's capacity for marine ecosystem conservation, research, and policy development while serving as a model for other small island communities facing similar challenges.

## 3 System Architecture and Design

### 3.1 Hardware

For hardware we developed 2 environmental monitoring systems. The air quality monitoring system and the water quality monitoring system.

Air Quality Monitoring System

| Component/Device | Use |
| --- | --- |
| Plantower PMS5003 | For reading particulate matter in PM(10.0,2.5 &1.0) µg/m3. |
| Sensirion SPS30 | For reading particulate matter in PM(10.0,2.5 &1.0) µg/m3. |
| Adafruit DHT31 | For reading relative humidity(%) and temperature(°C). |
| LilyGO TTGO T-SIM700G ESP32-WROVER | For computations, data processing, and remote controlling. |

## Water Quality Monitoring System

| Component/Device | Use |
|---|---|
| Atlas Scientific EZO Dissolved Oxygen Circuit | For reading dissolved oxygen in mg/L. |
| Atlas Scientific EZO RTD Temperature Circuit | For reading temperature(°C). |
| Atlas Scientific EZO Conductivity Circuit | For reading absolute conductivity in µS/cm |
| Atlas Scientific EZO pH Circuit | For reading pH |
| Atlas Scientific Basic EZO™ Inline Voltage Isolator | For separating electrical circuits from each other. |
| LilyGO TTGO T-SIM700G ESP32-WROVER | For computations, data processing, and remote controlling. |

## LilyGO TTGO T-SIM700G ESP32-WROVER (for both monitoring systems)

| Component/Device | Use |
|---|---|
| Built-in SIM7000G Module | For GSM communication through a cellular network. |
| Built-in Nano Sim Card slot | For SIM input. |
| Built-in SIM Antenna slot | For signal reception and transmission. |
| Built-in Li-ion/Li-Po battery charging circuit | For battery protection and provides solar charging interface. |
| Built-in 1x 18650 battery holder | For battery input, to store electrical energy. |
| Built-in solar panel connector 2p JST-PH | For connecting solar panels, enabling self-sustained power generation through solar energy |
| ESP32 chip (WROVER-B Module) (240MHz dual-core processor) | For data-processing, sleep cycles, and controlling the entire module. |
| 5V Solarpanel | To provide energy produced by the sun. |

The LilyGO TTGO T-SIM700G ESP32-WROVER development board served as an ideal choice for our environmental monitoring systems due to its comprehensive built-in components, minimizing the need for extensive soldering and simplifying our component list.  In the air quality monitoring system, we connected a DHT31 sensor and two PMS5003 sensors to the board. The DHT31 interfaces via I2C GPIO pins, while the PMS5003 sensors connect through UART GPIO pins. We also introduced extra headers connected to I2C GPIO pins to accommodate an SPS30 sensor for future developments.

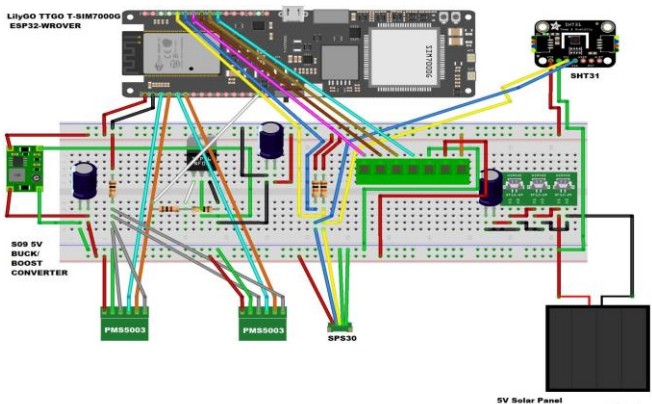

*Figure 1: Air quality monitoring system wiring diagram.*

For the water quality monitoring system, we integrated an array of sensors, including EZO Dissolved Oxygen, RTD Temperature, Conductivity, and PH sensors, all linked to the board via I2C GPIO pins. To ensure accurate readings, an EZO Voltage Isolator was incorporated to isolate each EZO sensor's circuit.

The wiring diagram for the water quality monitoring system can be seen below in figure 2. Must note that the water quality monitoring system uses the same components as the air quality monitoring system except for the PM sensors, SHT31 and buck boost converter, and in figure 2 only the distinct components are shown connected to the development board.

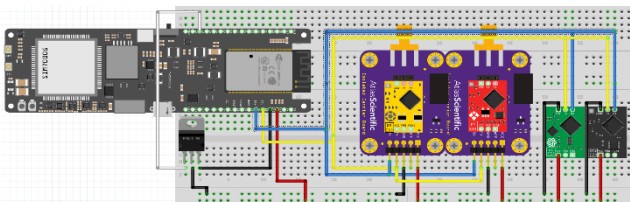

*Figure 2: Water quality system wiring diagram.*

To streamline our projects, we designed dedicated PCB boards for both the air and water quality monitoring systems. These PCB files are compatible with online PCB manufacturing services, simplifying the process of creating the necessary circuit boards.

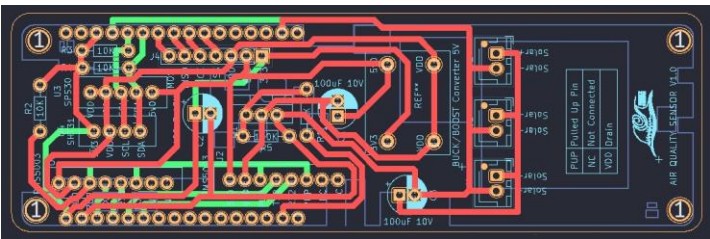

*Figure 3: PCB wiring diagram (Air Quality).*

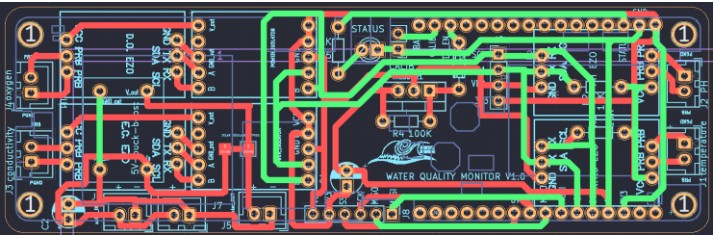

*Figure 4: PCB diagram (Water Quality).*

## 3.2   Software

### Air Quality and Water Quality Monitoring Systems

The air quality and water quality monitoring system follow a consistent software sequence. It begins with the ESP32 waking from sleep, creating sensor objects, and implementing a 10-minute watchdog timer (WDT) to allow for thorough checks in case of issues before resource cleanup.

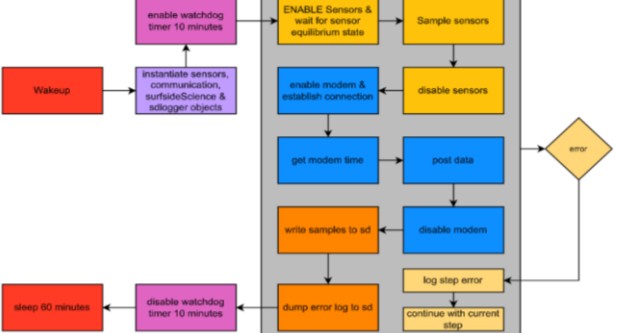

*Figure 5: Software sequence.*

In the Enable state, sensors are enabled and calibrated. Next, in the Sample state, data from sensors is sampled five times, followed by network connection establishment, data posting to the database, and modem deactivation. Simultaneously, sample data is written to an SD card, and any errors are logged

on the SD card. If errors occur in any state, they are recorded on the SD card. Before transitioning to the sleep state, the WDT is disabled. The ESP32 then enters a 1-hour sleep mode to conserve energy.

## Database

The initial database is built in PostgreSQL, with the PostGIS extension added later to allow for GIS data modeling. We decided to use PostgreSQL based on its support for GIS with the PostGIS extension, its broad market share and support (#4 on db-engines.com as of July 15th, 2022), and the developer's familiarity with PostgreSQL. Within the first few months, our priority is to design and build the database schemas as early as possible, as we require sensor data to be saved and queried as early as possible within the project. Later, we consider solutions that are simpler to deploy and manage, such as SQLite in combination with the SpatiaLite library extension. As for the platform for the interface between the database and the outside world, we chose Node.js using Feathers. The decision is based on the developer's familiarity with the systems, thus aiding the speed of development required for the first few months. Later, we will likely port the logic over to PHP as that is most easily available on hosting platforms. Having the code be usable as simply and as broadly as possible are some of the main goals for the project, and switching to PHP will accomplish that for the PHP framework, which will be determined before the switch to PHP.

## Data Portal

Surfside Science's Environmental Data Portal is your gateway to a wealth of environmental data collected by Brenchies Lab in Aruba. This comprehensive database covers a wide range of environmental aspects, including air quality, water quality, sea-floor conditions, vegetation, coastline data, and information about reef islands. This platform is meticulously designed to cater to the needs of scientists and policymakers, providing them with convenient access to essential coastal environmental data. To access the data, simply select your desired time frame, specify your area of interest, and choose the parameters relevant to that area. The system will then provide you with the available data for download in CSV format. Surfside Science continuously updates and enhances the portal, recently adding new features like Vite.config.js for managing pages and dedicated upload pages with their own Vue files. It's a dynamic resource for anyone interested in the environmental landscape of Aruba.

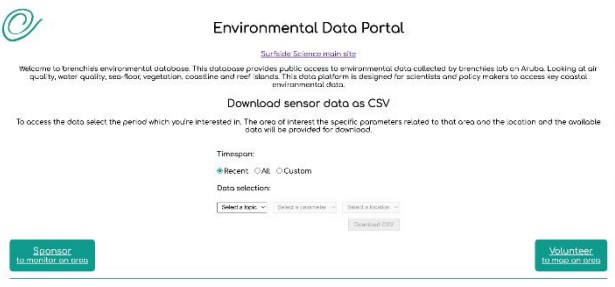
*Figure 6: Data Portal.*

## AI mapping

Surfside Science employs a multifaceted approach to seafloor mapping that combines innovative technology and cutting-edge machine learning techniques. The data collection process begins with underwater imagery, captured by Kayaks equipped with GoPro cameras set to timelapse mode. These cameras snap images every half second, approximately every 1.5 meters, ensuring comprehensive coverage. To precisely geolocate these images, GPS logging during the excursions is conducted, and GPS tagging using EXIFTOOL further enhances the data. These geolocated images are then uploaded to a centralized database.

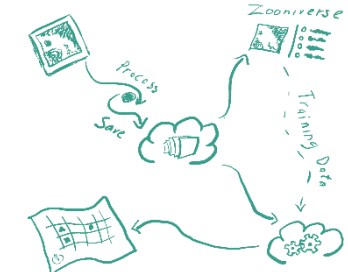
*Figure 7: Workflow Seafloor Mapping.*

The next step involves machine learning, where a selection of these images is uploaded to Zooniverse for labeling. The labeled images are then used to train a machine learning model created with TensorFlow and Keras, coded in Python. The model is trained to classify the classes: *coral or sponges, seagrass or seaweed, sand* and *rocks, and rubble.* The kayak-derived images created a non-comprehensive map. Conversely, the satellite imagery created a comprehensive map, based on training data selected from Google Earth engine. Ongoing deliberations center on the prospect of synergistically amalgamating these divergent methodologies. This entails the utilization of AI-classified imagery to inform the training of the satellite-based classifier, thereby fostering an advanced and well-informed approach to comprehensive maps. It is essential to emphasize that this integrative endeavor remains in the preliminary stages of exploration and has not yet been empirically examined.

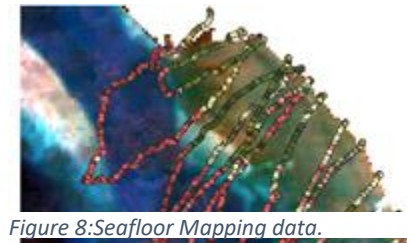

*Figure 8:Seafloor Mapping data.*

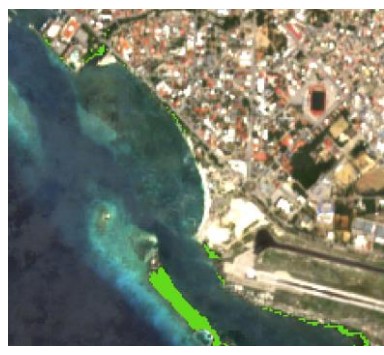

## Coastal Mapping Using GIS Techniques and Satellite Imagery

At Surfside Science, our approach to coastal mapping in the Surfside Beach region of Aruba integrates local knowledge with Geographic Information Systems (GIS) techniques applied to satellite imagery through Google Earth Engine. This innovative methodology allows us to monitor changes in the coastline, assess the dimensions of reef islands, and comprehensively evaluate beachside vegetation, all with the aim of understanding the coastal dynamics of this unique environment.

### Satellite Imagery and Google Earth Engine:

A cornerstone of our coastal mapping methodology is the utilization of Google Earth Engine, a robust platform that empowers us to harness open satellite imagery datasets. This powerful tool enables us to perform detailed analysis and mapping efficiently. The combination of GIS techniques and Google Earth Engine's capabilities positions us to explore changes in the coastal landscape, including its dynamic shoreline.

### Coastal Vegetation Mapping with NDVI:

In our quest to map coastal vegetation, we employ the Normalized Difference Vegetation Index (NDVI). This index, derived from satellite imagery, serves as a valuable indicator of the health and density of vegetation along the coastline. NDVI calculations are based on the reflectance of near-infrared (NIR) and red light, with healthy vegetation reflecting more NIR and absorbing more red light. This analysis enables us to produce a precise map of coastal vegetation, helping us better understand the composition and condition of this critical ecosystem.

### Reef Islands and Coastline Delineation with mNDWI:

To further our understanding of the coastal environment, we employ the Normalized Difference Water Index (NDWI). This index assists in the accurate delineation of reef islands and the identification of coastline boundaries. Like NDVI, the NDWI leverages satellite imagery, this time assessing the reflectance of green and near-infrared (NIR) light. This innovative approach facilitates a comprehensive understanding of coastal dynamics, allowing us to analyze changes in the size and extent of these vital landforms.

### Shallow Marine Habitat Mapping:

As part of our coastal mapping strategy, we delve into the mapping of shallow marine habitats, a critical ecological zone. This process employs supervised classification techniques to identify and categorize these areas. Through the integration of Python, we automate the periodic generation of maps, enabling real-time monitoring and analysis. Our adaptable code can be easily customized to accommodate various geographic regions and satellite datasets, making our approach versatile and scalable.

# 4 Addressing Challenges

## Air Quality

During the development of the initial prototype for the air quality monitoring system, we encountered a challenge when the SPS30 sensor proved to be defective. To prevent project delays, we proceeded without the SPS30 and instead incorporated two PMS5003 sensors for particulate monitoring. Nevertheless, we designed a slot in the system for later integration of the SPS30 once new orders arrived. Additionally, since our initial prototype was constructed on a self-milled PCB board with subpar copper quality, soldering had to be performed carefully almost with surgical precision.

## Water Quality

Much like the air quality monitoring system, the initial water quality prototype encountered soldering challenges due to the use of the same lower-quality PCB board. Furthermore, the sensors required delicate handling, as splicing or cutting the sensor wires and soldering them would compromise accuracy. Additionally, the project faced a significant challenge in designing a suitable water-tight housing for the system, given the corrosive nature of seawater. This necessitated the use of anti-biofouling materials for effective protection.

## Coastal Mapping

During the development of our coastal mapping system, one of the challenges we encountered was related to the validation of vegetated areas. Regrettably, due to the time-intensive efforts required to establish an accurate coastline, we faced limitations in allocating sufficient time for the comprehensive validation of vegetated regions. To address this challenge, we propose an alternative method that considers the size of the vegetated area rather than its precise location. This approach aims to circumvent the challenges experienced during coastal validation, ensuring a more effective and time-efficient validation process for vegetated areas in future iterations of our system. By prioritizing area size over precise location, we can streamline the validation process and enhance the accuracy of our vegetated area mapping.

## Surfside Mapping

We faced a challenge regarding the accuracy of our seafloor mapping system. To address this, we recognized the need for more testing data, which could improve the system's robustness. Additionally, we identified challenges related to GPS-based errors affecting both AI and ground-truthing data. To overcome these challenges, we sought ways to reduce the impact of these errors in our mapping process.

The validation accuracy was lower than desired due to having too few data points within 5 meters of ground observations. We addressed this by exploring methods to increase the density of data points and improve the accuracy of ground observations, potentially by refining the data collection process.

# 5 Performance Evaluation and Testing Results

## Air Quality

A test was done using our Air Quality Monitoring Sensor 1 (AQMS1) prototype alongside established reference meters to measure air quality in real-world conditions. These reference meters, including popular models like GAIA, provide high accuracy and reliability. While AQMS1 aligns closely with GAIA, it shows slight differences compared to high-end meters but is considered adequate for general air quality monitoring. AQMS1's PM values are affected by humidity and temperature, but they still serve as a good indicator of air quality, offering insights into trends and public health hazards while enabling the calculation of the WHO's Air Quality Index. In Aruba, where high-quality data is lacking, low-cost air quality devices like AQMS1 provide a cost-effective solution.

## Water Quality

A test was done using our Water Quality Monitoring System 1 prototype (WQMS1) alongside reference meters. Challenges were encountered in finding suitable local reference meters, leading to validation using instruments like the Hanna Instruments HI 98194 and SeaSun CTD48. Ultimately, the team acquired the HI 98194 for local validation.

Test results showed significant differences, with limited overlap and higher variability in the WQMS1 data compared to the SeaSun reference. Minor temperature differences of 0.2°C were observed. Correctable offset issues were identified for temperature, electrical conductivity (EC), and dissolved oxygen (DO), underscoring the importance of sensor calibration despite manufacturer claims to the contrary. However, calibration results for pH and EC were mixed and somewhat confusing.

The corrected WQMS1 devices have the potential to indicate ecologically disturbing events, such as coral bleaching and eutrophication, as well as changes in coastal habitats.

## Coastal Mapping

Pre-processing of satellite imagery to remove cloudy pixels and combine data over a date range of one month was performed to form a reliable base image for analysis. The maps that were created from satellite image analysis were assessed by a variety of techniques depending on the type of map and the suitability of available methods. Seafloor maps were compared against sea-truthing data collected by divers and classified by a marine scientist. These were validated to have an accuracy of 60.7%. These maps were also compared against available existing seafloor maps of the area and had a higher accuracy than maps produced by the Allen Coral Atlas (2022), and the Carmabi Foundation (2020). GPS devices planned for use as coastline and reef island validation had accuracy issues with discrepancies of 10 to 15m, so they were not possible to use. Instead, coastline maps were compared between dates and showed consistency within tidal ranges. Reef island maps were compared against manually traced outlines of high-resolution imagery and were shown to accurately count the number of islands and to overestimate the area by 10%. Vegetation maps were compared against field observations to confirm the presence or absence of mangroves, but boundary accuracy was not calculated due to the inaccuracy of available GPS equipment.

## Surfside Mapping

*Figure 9:Level of correctness classification of seafloor data at Surfside Bay Aruba.*

**Testing Accuracy**: Our system demonstrated a commendable testing accuracy of 81.0%. This performance was evaluated through a rigorous testing process, which involved assessing the system's ability to classify seafloor data accurately. Despite the high accuracy, we acknowledged the need for more testing data to further validate our results.

**Validation Accuracy**: In contrast, the validation accuracy stood at 20.0%. We evaluated the system's performance using ground observations, but the limited number of data points within 5 meters posed a challenge. This result underscored the need for improving the accuracy of ground observations and addressing potential GPS-based errors affecting both the AI model and the ground-truthing process.

**Recommendations for Improvement:** To enhance our system's performance, we proposed several key improvements. These include redefining classification labels, implementing a 70%:30% training-to-testing data ratio, experimenting with different AI threshold values, and validating AI image classification outputs. These enhancements aim to increase the reliability and accuracy of our seafloor mapping process, ensuring it meets the high standards required for ecological preservation and resource management.

# 6 Concluding Remarks and Avenues for Future Work

In conclusion, Surfside Science is a project dedicated to improving environmental data collection, particularly in small island nations like Aruba. It has successfully implemented monitoring stations for air and water quality and developed a marine mapping system, contributing to a better understanding of coastal and marine ecosystems. The open-access data portal ensures information is readily available to scientists and policymakers.

While the project has encountered challenges, mainly for the water quality monitoring system for calibration and data validation, its commitment to improvement and adaptability are strengths that will contribute to its long-term success. This project is still fresh and ever evolving, and the Surfside science team is committed to making more improvements to the project.

Surfside Science's focus on open-source knowledge sharing and replicable methods makes it a valuable model for enhancing environmental data accessibility and empowering communities. It has the potential to create a positive impact not only in Aruba but also in similar regions worldwide, addressing crucial environmental challenges.

# 7 Availability

Please provide the URLs for the software repository and the URL for the video where you demonstrate the system in this section.

- Seafloor Mapping Webpage: https://science.brenchies.com/en/seafloor-mapping/
- Zooniverse webpage: https://www.zooniverse.org/projects/brenchies/aruba-seafloor-mapping
- GitHub repo (Air & Water Quality): https://github.com/brenchies/SurfsideSensors
- GitHub repo Coastal mapping: https://github.com/brenchies/surfsideGEE
- Surfside Science webpage: https://science.brenchies.com/en/
- Data portal: https://gitlab.com/Blind238/surfside-science-db/