# OpenReview forum: "Surfside Science:  Empowering Communities with Environmental Monitoring Innovations"
_helsinki.fi/ESPC/2023/Competition — ESPC 2023 LongPresentation_

### Official Review · Reviewer_9RtN · 2023-11-14

**Rating:** 4
**Confidence:** 4

**Summary:**

The project aims at offering a holistic and comprehensive solution for monitoring the environment. The system can be leveraged for longitudinal studies. Furthermore, the system is targeted to be easy to deploy and use in small independent island states.

**Strengths:**

- The system has been deployed and used in real-life studies.
- The evaluation highlights the strengths and weakness of the system. Furthermore, it includes a critical analysis of the quality of data collected by the sensors, and it is validated against ground truth.
- The system also has an API that can be leveraged by the community to further analyze the data.
- The hardware design is detailed in the report, and the URLs point to the open source components. The source code is documented using the best practices and the instructions and README files are clear.
- The tradeoffs and the choices made are detailed. For instance, the reasons behind the choice of postgresql and node.js has been provided in sufficient detail.
- The challenges and the approaches to address the challenges have also been presented in sufficient detail.

**Weaknesses:**

None that are worth mentioning. Here are some suggestions to further improve the work (if not done already)
- Some micro-benchmarks to quantify the components that consume the most energy.
- Some details on how you plan to address GPS errors would further strengthen the work done.

---

### Official Review · Reviewer_daTK · 2023-11-16

**Rating:** 4
**Confidence:** 2

**Summary:**

Report presents "Surfside Science" project that aims to provide methodologies to enhance accessibility to data collection systems for Small Island Developing States (SIDS), by creating affordable environmental open-source monitoring prototypes, accessible to the public through the project website. Currently, project includes air and water quality monitoring, seafloor mapping, and coastal change analysis.

**Strengths:**

Very ambitious project, involving a number of experts. Interesting and very important activity.
- Solutions are fully designed, implemented, and deployed.
- Data is available through the data portal.

**Weaknesses:**

- It would be interesting to dive deeper into implemented models, no actual details is given in the report.
- Maybe I missed from the report, but how the labelling of the images was done?
- What GIS techniques used?
- For performance, would be nice to see some results in terms of graphs also.
Overall, very interesting and comprehensive work. I would assume that all the details was hard to put into the report, given the page limits.

---

### Official Review · Reviewer_8iYC · 2023-11-17

**Rating:** 4
**Confidence:** 4

**Summary:**

This project is part of an EU project "Surfside project". In this part, two environmental factors are monitored: air and water quality. For air quality monitoring, the key parameters measured are PM10.0, 2.5, and 1.0, humidity, and temperature using sensors such as SPS30, PMS5003, and SHT31. For water quality monitoring, the parameters considered are pH, dissolved oxygen, temperature, and electrical conductivity. In both the cases, the data is transmitted to a central database via GSM communication. The systems are energy-efficient and designed for robustness and a long lifespan, estimated to last for several years of data collection.

**Strengths:**

1. The presentation is very good and includes motivation, block diagram, circuit diagram, actual deployment, challenges faced, results.
2. The project is well executed, which is not surprising since this is part of a bigger EU project.

**Weaknesses:**

1. The presentation is confusing in terms of scope. Although the openReview page says that only two parts of air and water are part of this competition, but the pdf is for the whole of the project.
2. It is not clear how much data is collected and when.
3. Calibration details are not clear.

---

### Official Review · Reviewer_2Qak · 2023-11-18

**Rating:** 4
**Confidence:** 3

**Summary:**

This project aims to develop methodologies to monitor critical environmental factors, such as air quality, water quality, coastline monitoring, and seafloor mapping.
They have established data collection and monitoring stations for aquatic life, air quality, marine mapping, and validating GIS data.
They presented hardware design for the two systems they have developed -- Air Quality Monitoring System and Water Quality Monitoring system. They have developed a backend database, a nice user portal, AI mapping techniques to generate seafloor map that rely on underwater images and machine learning.
They have also done performance evaluation for the system they have developed.

**Strengths:**

Their idea is novel, presented clearly and comprehensively, and has a high environment impact.
Entire workflow of the system from hardware to software is implemented in its entirety.
This system has wide range of applications as they have mentioned in their report -- they can monitor changes in the coastline, reef islands, beachside vegetation, marine habitats.
The evaluation results show that the accuracy is pretty close to reference meters.
Their tools are live, and code is open-source.

**Weaknesses:**

Better image resolution in the report would have helped.

---

### Official Review · Reviewer_AAe4 · 2023-11-18

**Rating:** 2
**Confidence:** 4

**Summary:**

This project “Surfside Science” aims to develop and validate methodologies that enhance accessibility to data collection systems for Small Island Developing States (SIDS), primarily by creating affordable and accurate environmental monitoring prototypes and making them open-source, accessible to the public through our project website. The report and the presentations show the schematic of sensor deployment but lack to show how they are deployed and how the data is collected. The project aims to collect data in different environments such as air, soil, water, vegetation, etc., but it is not clear how with the usage of the low-cost sensors they do it. However, they mention they use satellite data but the way they access it is unclear. The project also presents https://data.brenchies.com/ a database that provides data for the public, which is nice and appreciated.

**Strengths:**

The idea of collecting environmental data from Small Island Developing States and provide publicly available datasets.

**Weaknesses:**

It is not clear if the project is addressing the development and deployment of low-cost sensors for different environments or using satellite data from a specific environment. Or, maybe the whole idea of Surfside Science that aims to tackle many areas of the environmental concerns. For this student competition, I wished to see a contribution that focuses on only one specific aspect.

---

### Official Review · Reviewer_Tw4E · 2023-11-20

**Rating:** 4
**Confidence:** 4

**Summary:**

This project developed and validated methodologies that enhance accessibility to data collection systems for Small Island Developing States (SIDS), primarily by creating affordable and accurate environmental monitoring prototypes and making them open-source, accessible to the public through our project website. The prototypes were designed to be replicable, by individuals and communities where environmental monitoring efforts are currently limited. The aim is to empower people to become environmental scientists, providing them with the tools and knowledge to monitor and address climate and environmental issues in their regions.

**Strengths:**

This is an excellent submission.  The Caribbean islands are highly vulnerable to the adverse effects of climate change, and the Surfside Science data is essential for understanding their unique environmental challenge. The project team developed monitoring methods to measure air quality, water quality, coastline monitoring, and seafloor mapping. The methods provided real-time, reliable data and have been operational for over a year. The report introduction is well written the goals are concrete. The system hardware is very interesting particularly the combination of air quality and water quality sensors. Good software. database and portal description. Coastal and sea floor mapping using Kayaks and satellite imagery, and applying AI are highlights of this work. The team calibrates the low cost devices and then evaluates the performance of the devices  over time, and give accuracy recommendations. The video is very information and data are available in GItHub.

**Weaknesses:**

The report could have described how the different environmental monitoring methods come together.  A diagram connecting air quality, sea water quality, coastal and seafloor mapping as one interrelated system would have been very informative.  A diagram or pictures of the combined air quality and water quality sensor was missing. Also how were this multifunctional device are used in practice. There are not analytics time series figures showing the performance of the air quality/water water sensor over time. . But overall, inspiring work..